# How COVID-19 Phases Have Impacted Psychiatric Risk: A Retrospective Study in an Emergency Care Unit for Adolescents

**DOI:** 10.3390/children9121921

**Published:** 2022-12-08

**Authors:** Maria Mucci, Francesca Lenzi, Giulia Maria D’Acunto, Marisa Gazzillo, Ilaria Accorinti, Silvia Boldrini, Giacomo Distefano, Francesca Falcone, Beatrice Fossati, Roberto Giurdanella Annina, Silvia Paese, Carmen Salluce, Irene Troiano, Cinzia Fratoni, Deborah Fabiani, Francesca Liboni, Gabriele Masi

**Affiliations:** 1IRCCS Stella Maris, Scientific Institute of Child Neurology and Psychiatry, Calambrone, 56025 Pisa, Italy; 2Department of Clinical and Experimental Medicine, University of Pisa, 56126 Pisa, Italy

**Keywords:** COVID 19, psychiatric emergency unit, adolescence, suicidality, non-suicidal self-injuries, traumatic adverse events

## Abstract

Dramatic events during the COVID-19 pandemic have acutely impacted the psychosocial environment worldwide, with negative implications for mental health, particularly for more vulnerable children and adolescents with severe psychiatric illnesses. Some data suggest that the pandemic waves may have produced different psychopathological consequences, further worsening in the second phase of the pandemic, compared to those in the first lockdown, soon after March 2020. To test the hypothesis of a further worsening of psychiatric consequences of COVID-19 in the second lockdown compared to the first lockdown, we focused our analysis on a consecutive sample of youth referred to a psychiatric emergency unit for acute mental disorders in the time period between March 2019–March 2021. The sample, consisting of 241 subjects (123 males and 118 females, ranging in age from 11 to 17 years), was divided into three groups: Pre-Lockdown Group (PLG, 115 patients); First Lockdown Group (FLG, 65 patients); and Second Lockdown Group (SLG, 61 patients). Patients in the SLG presented more frequently with non-suicidal self-injuries (NSSIs), suicidal ideation, and suicidal behavior, while no significant differences in self-harm were found between PLG and FLG. Eating disorders were more frequent in both the FLG and SLG, compared to the PLG, while sleep problems were higher only in the SLG. Furthermore, patients in the SLG presented with more frequent psychological maltreatments and neglect, as well as with psychiatric disorders in the parents. Adverse traumatic experiences and internalizing disorders were significantly associated with an increased risk of suicidality. Intellectual disability was less represented from the PLG to SLG, and similarly, the rate of ADHD was lower in the SLG. No differences were found for the other psychiatric diagnoses. This information may be helpful for a better understanding and management of adolescents with severe emotional and behavioral disorders after the exposure to long-lasting collective traumas.

## 1. Introduction

Dramatic events during the COVID-19 pandemic, such as loss of lives, isolation, contact restrictions, disruption of daily rhythms (particularly eating and sleeping routines), and economic shutdown have acutely impacted the psychosocial environment worldwide. This collective trauma led to considerable threats to mental health, particularly for more vulnerable children and adolescents [1,2,3]. In an extensive review including studies on children and adolescents worldwide [4], anxiety and depression symptoms were commonly reported, as well as irritability and anger. In this study, protective factors included family communication, social supports, and appropriate play and leisure, while among risk factors, adolescence, female sex, and above all, previous mental disorders before the lockdown, had a key role [4]. This negative impact was lower for milder forms of psychopathology, and some improvements actually occurred in these patients, particularly in the first phases of the pandemic, when a pause on the demands of in-person schooling (peer interactions, sensory over-stimulation, etc.), as well as increased access to supportive parents, may have reduced the perceived stress [1,4]. On the contrary, individuals with more severe pre-existing psychiatric illnesses were especially vulnerable to the consequences of COVID-19 [5,6,7], particularly suicide-related ideation and behavior [8,9,10]. Children and adolescents with exposure to previous or current adverse events and traumatic experiences were also especially vulnerable for consequences of the COVID-19 crisis [11], particularly physical and psychological violence within families [5,12,13].

Different pandemic “waves,” starting from March 2020, occurred, with different emotional impacts on the general population, which were more pronounced in vulnerable youth. While the first lockdown, soon after March 2020, was associated with a mix of fear and optimism, as well as a spirit of reaction and hope for a short duration of the crisis (“Everything will be all right”), the second lockdown, after fall of 2020 and during winter of 2021, was associated with fallen hopes and increased fears for a long duration of the crisis. Furthermore, in the second lockdown, more flexible, but sometimes inconsistent adjustment to the children’ rhythms were introduced, while parents went back to work, and youth were more frequently alone at home [14,15]. Analyses regarding the effects of these changing phenomena on vulnerable adolescents referred to psychiatric services are lacking. The impact of these COVID 19-related changes to daily life activities and the differential effects of these collective feelings on vulnerable adolescents referred to psychiatric services are still unknown.

Based on these considerations, we focused our attention on the most severe psychiatrically ill youth, that is, those who required hospitalization in a psychiatric emergency unit for children and adolescents. To explore this issue, three different time periods were analyzed: the pre-lockdown period, which preceded the pandemic (March 2019 to March 2020), the first lockdown period (March 2020 to October 2020), and the second lockdown period (October 2020 to March 2021) [2,16]. We hypothesized that the most severe, vulnerable, and sensitive patients may have more clearly expressed a greater, cumulative impact during the second lockdown, compared to the pre-lockdown and first lockdown periods. More specifically, we hypothesized that during the second lockdown, global severity and functional impairment of the referred patients was more severe. We also hypothesized that specific clinical conditions, such as mood disorders (depression and bipolar disorder), anxiety disorders, disruptive-impulse control and conduct disorder behavior, suicidal ideation and behavior, NSSI, substance use disorder, eating disorders, and sleep disorders were more frequently represented. We hypothesized that traumatic environmental experiences may have occurred more frequently during the second lockdown, compared to the first lockdown and the pre-COVID-19 period. Finally, we hypothesized that adverse traumatic experiences and internalizing disorders may increase the risk of suicidality.

If confirmed, these hypotheses may provide helpful insights for the management of adolescents with severe emotional and behavioral disorders after the exposure to long-lasting collective traumas.

## 2. Materials and Methods

### 2.1. Sample

This is a retrospective study based on a consecutive clinical database of all the 241 adolescents, 123 males (51% of the sample) and 118 females, referred as inpatients to the psychiatric emergency unit of our hospital with region-wide catchment (Tuscany) between March 2019 and March 2021 for acute emotional and/or behavioral conditions (i.e., agitation and aggression, suicidal or non-suicidal self-harm, isolation, suicidal risk, psychotic symptoms, etc.).

The sample was divided into three groups according to the time of the referral (see Table 1):− Pre-Lockdown Group (PLG) (9 March 2019–8 March 2020): 115 patients, 63 males (54.3%) and 52 females (44.8%); mean age of 14.0 ± 2.7 years;− First Lockdown Group (FLG) (9 March 2020 to 10 October 2020): 65 patients, 33 males (50.8%) and 32 females (49.2%); mean age of 13.7 ± 2.9 years;− Second Lockdown Group (SLG) (11 October 2020 to 30 March 2021): 61 patients, 27 males (44.3%) and 34 females (55.7%); mean age 14.09 ± 2.0 years).

### 2.2. Measures

All the patients referred to our unit were routinely assessed with the same diagnostic protocol, aimed to assess possible risk for suicidality, according to a preventative research project, since 2014.

− Schedule for Affective Disorders and Schizophrenia for School-Age Children-Present and Lifetime Version (K-SADS-PL) [17], a semi-structured interview administered by trained child psychiatrists to patients and parent(s), to obtain categorical diagnoses, according to the DSM-5 criteria;− Child Global Assessment Scale (C-GAS) [18], to assess the functional impairment, with a score ranging from 0 (needs constant supervision) to 100 (superior functioning);− Child Behavior Checklist (CBCL) [19], a 118-item scale, completed by parents, to obtain a dimensional diagnosis of psychopathology, clustered in two broad-band scores—Internalizing Problems and Externalizing Problems—and a Total Problem Score, along with 8 different syndromes scales (withdrawal, somatic complaints, anxiety/depression, social problems, thought problems, attention problems, rule-breaking behavior, aggressive behavior);− Columbia–Suicide Severity Rating Scale (C-SSRS) [20] for the assessment of the suicidal ideation and behavior. Suicidal ideation in the sample is defined by a score 3 or above.− Assessment of the DSM-5 categorical diagnostic criteria for non-suicidal self-injury (NSSI), that is, NSSI on at least 5 days within the past year, suicidal ideation absent or low (score below 3 at the C-SSRS), and no previous suicide attempts.

Medications at admission and at discharge (antidepressants, mood stabilizer, benzodiazepines, and antipsychotic drugs) were also reported.

Historical information was retrospectively collected using an unstructured checklist exploring the presence of parental separation/divorce, bullying, second-generation immigration, familial psychiatric disorders (namely, mood disorders), or familial attempted or completed suicides. Furthermore, the presence of lifetime traumatic life experiences was assessed with a checklist, by disentangling exposure to direct adverse childhood experiences (ACE), such as childhood maltreatment, physical or sexual abuse, psychological abuse, and indirect ACE, such as emotional neglect, familial conflicts, or family violence [21].

All procedures were in accordance with the ethical principles of the Declaration of Helsinki, and approved by the Ethics Committee of Hospital Meyer, Florence Italy, protocol code 0001507, 2014.

### 2.3. Statistical Analyses

The statistical analyses were conducted with the Statistical Package for Social Science (SPSS) 19. Descriptive analyses were used to describe demographic and clinical characteristics of the whole sample. Chi-square analyses were performed on categorical variables, and two-ways ANOVAs on continuous variables. Regarding the comparisons between the three groups, for dichotomous variables, z test were conducted. P values were based on two-tailed tests, with α = 0.05. A regression model was used to identify predictors of suicidality; a binary logistic regression (stepwise Wald method) was performed to evidence risk or protective factors for suicidality, and considering demographic parameters (group, age, and gender), the internalizing CBCL subscale and traumatic predictors were used as independent variables (Nagelkerke R square index).

## 3. Results

The patients included in the study presented severe functional impairment according to the C-GAS scores (28.0 ± 7.8 in PLG, 27.5 ± 7.3 in FLG, 29.8 ± 6.7 in SLG), without significant differences among groups.

Patients in the SLG presented more frequently NSSI, suicidal ideation, and suicidal behaviors, but not substance use disorder, compared to both PLG and FLG, while no significant differences were found between PLG and FLG. Eating disorders were more frequent in both FLG and SLG, compared to the PLG, while sleep problems were higher in the SLG. The frequency of referrals for intellectual disability decreased from PLG to SLG, and similarly, attention deficit hyperactivity disorder (ADHD) was lower in the SLG. No differences were found for all the other categorical diagnoses (Table 1).

The differences between direct and indirect ACEs are shown in Table 2 Patients in the SLG presented more frequently than those in the PLG with psychological maltreatments and neglect, as well as psychiatric disorders in the parents, while no differences were found for sexual abuse or physical maltreatment, witnessed violence, alcoholism, family member jailed, and uniparental family (Table 3).

No differences between groups were found in dimensional psychopathology, assessed with the CBCL (parent version) (Table 3).

No differences were found in any class of medications at admission (antidepressants, mood stabilizer, benzodiazepines, and antipsychotic drugs). At discharge, only antidepressants were prescribed more frequently in the SLG compared to the PLG (31.7% and 14.7%, respectively.

A binary logistic regression aimed to analyze the risk factors for suicide attempts, including demographic parameters (group, age and gender), internalizing CBCL subscale, and traumatic predictors as independent variables (Nagelkerke R square index) showed that the final model accounted for 53.1% explanatory risk factor for suicide, with all independent variables listed in Table 4.

## 4. Discussion

The aim of this study was to evaluate the putative differential impact of the first and second lockdowns in COVID-19 in a sample of children and adolescents referred to a psychiatric emergency unit for children and adolescents.

In contrast to our first hypothesis, patients in the second lockdown were not more severely impaired than those in the first lockdown or in the pre-COVID-19 period. Our patients, seeking urgent admission in an emergency psychiatric unit, were likely so severely impaired, as evident from the C-CAS scores in all the three groups, that a lockdown effect was not evident.

Consistently, with the second hypothesis, non-suicidal self-injuries (NSSI), along with suicidal ideation and behaviors, significantly increased from PLG-FLG to the SLG. Significantly, no substantial differences were found between the PLG and the FLG. A three-fold increase was registered for suicide attempts from the FLG to the SLG. This finding was associated with a two-fold increased use of antidepressants in the SLG, compared to the PLG. The findings on NSSI are in line with those of Hoekstra [22], Du et al. [23], and Ougrin et al. [9]. Moreover, the increase in suicidality in the SLG is consistent with epidemiological data from the USA, with emergency department visits for suicidal ideation and behavior decreasing in the first months of the pandemic, then increasing during summer 2020, further rising through the winter 2021, especially among adolescent girls (up by 50% compared to 2019) [24]. Similarly, a Spanish study compared data from March 2020 to March 2021, and found a strong increase in suicide attempts among adolescents, particularly in girls (up to a 195% increase) after the end of confinement measures in September 2020 [25]. Interestingly, another Spanish study, including not only adolescents, but the general population, reported a decrease in emergency department visits for suicide attempts or persistent suicidal ideation in the first period soon after the onset of the COVID-19 crisis, compared to the previous two years [26]. Consistently, in a cohort sample of 234 adolescents admitted for persistent suicide ideation, Mourouvaye et al. [27] found a significant decrease in the incidence of admissions for suicide behavior during the period of March-May 2020 (corresponding to the first wave), without information about the second wave. These findings may suggest that soon after the onset of the COVID-19 emergency, suicidality failed to increase, compared to the following periods. A non-alternative explanation is that the first phase of COVID-19 was characterized by decreased hospital admission rates, and generally reduced help-seeking, beyond reasons related to COVID-19 infections, and this phenomenon may explain the apparent decrease in referrals for suicidality [27].

It has been hypothesized that the increase in internalized symptoms during the pandemic may have determined the increase in suicidal behaviors over the long term [25,28,29]. Of note, in contrast to our hypotheses, no differences were found in our sample regarding rates of depression and anxiety, nor for bipolar disorder and disruptive behavior disorders (oppositional defiant disorder and/or conduct disorder). A dimensional approach to psychopathology (using the CBCL) consistently failed to find a higher rate of internalizing and externalizing problems. Considering the higher rates of NSSI and suicidality reported above, especially during the second lockdown, it may be argued that this strong increase in self-harm ideation and behavior may be specific, and not secondary to an increase in mood, anxiety, or impulse-control disorders, both in categorical and dimensional terms. Our findings, limited to a context of acute psychiatric referrals, suggest that the rates of most severe forms of affective and behavioral disorders were not significantly affected by the pandemic. This is inconsistently reported in the literature. Early international research suggested that the COVID-19 pandemic was associated with more elevated rates of psychiatric disorders in adolescents, particularly depression and anxiety [30]. On the contrary, another study [31] comparing 100 adolescents in the pre-pandemic group and 134 in the pandemic group, showed that neither being in the pandemic phase, nor experiencing changes in daily activity due to the pandemic, was associated with higher depression or anxiety.

Eating disorders strongly increased from the pre-lockdown and the first lockdown, with a further, strong increase in the second lockdown. This is consistent with other studies reporting on a higher frequency of acute care visits for pediatric eating disorders after the onset of the COVID-19 pandemic, and during the first 10 months of the pandemic [32]. Growing stress and anxiety, social isolation, disruption of routines, and increased free time may have triggered disordered eating behaviors, worsened pre-existing problematic behaviors, or disrupted previous treatments [2,33]. Not surprisingly, in these studies, individuals with eating disorders reported increased social isolation, rumination about eating, feelings of anxiety and depression, and decreased feelings of control and social support during the COVID-19 pandemic [4,34].

The higher prevalence of sleep problems in SLG with respect to both PLG and FLG is not surprising, considering the disruption of sleep-wake cycles during the COVID-19 pandemic. Panchal et al. [4] reported that rates of children with sleep disorders (both initiating and maintaining sleep) increased from 40 to 62% during lockdown. Consistently, Panda et al. [35] reported an increase of up to 21.3% of youth with sleep disturbance. Finally, Bera et al. [2] underlined the extensive use of screens, particularly during the night, as associated with disrupted sleep rhythms.

Regarding the role of traumatic experiences, consistent with the hypothesis, higher rates of psychological maltreatment and neglect, and increased parental psychiatric pathology were found in the SLG, compared to the PLG. The literature underlines an increase in child maltreatment and neglect during the pandemic [36,37]. Economic problems (unemployment and wage cuts), difficult family management related to school closures and resuming work, limitations in social contacts, and the cancellation of out-of-home leisure time activities may have increased family conflicts. This can severely affect parenting and—in the worst cases—erupt in physical and psychological violence withing families [5,12,13]. Fegert et al. [5] stressed the unmet need for evidence-based interventions aimed at improving adolescents’ resilience in regards to external contingencies and interpersonal coping skills, as well as parents’ skills in detecting their own offspring’s risk factors of uneasiness The training of clinicians and the implementation of psychiatric services according to a “trauma-informed-care” approach may be crucial for a timely and effective evaluation and treatment, especially during acute psychiatric hospitalizations [38]. Some principles of this approach (i.e., reduce re-traumatization, support the development of specific skills and healthy short- and long-term coping mechanisms) may further improve treatment strategies and promote new organizational models of highly competent and specialized psychiatric emergency services.

Two additional findings should be underlined. The first is that ADHD was more represented in the FLG, compared to the SLG. This finding may be explained as a decrease in help-seeking in less severe clinical conditions, such as ADHD, compared to self-harm behaviors, in an emergency setting, since parents may manage moderate disruptive behavior by themselves, instead of recurring to hospitalization [27]. Data from the literature are controversial, as some authors found a global worsening in general well-being in ADHD children, including oppositional/defiant attitudes, emotional outbursts, sleep problems, and anxiety [4]. Other studies have reported an improvement in ADHD symptoms across the pandemic period. In a survey for parents of children and adolescents with ADHD, Bobo et al. [14] showed an improvement in anxiety in children with ADHD, along with improved self-esteem as one of the main topics addressed by parents. Consistently, Bera et al. [2] also reported such improvements, related to fewer school-related strains and more flexible schedules.

Another relevant finding was the lower rate of intellectual disability in the SLG compared to PLG. As discussed by Mourouvaye et al. [27], this finding might result from reduced help-seeking and decreased hospital admissions of disabled patients, often labeled as the “Cinderella” of psychiatry, compared to acute patients without intellectual disability. This phenomenon may suggest that in time of crisis, people with intellectual disability may have been the least supported patients.

To deepen possible elements associated with an increased risk of sever suicide ideation and attempts, internalizing disorder and adverse childhood experiences emerged as significant risk factors, consistent with the literature data and our previous findings [10,39,40]. Regarding age and gender differences, our findings are consistent with prior studies, which showed that suicide attempts are more common in adolescent females. Our findings strongly support the literature regarding the role of adverse childhood experiences, particularly parental psychiatric disorder.

Our findings should be interpreted cautiously, in light of the main limitation, that is the referral bias represented by very severely patients admitted to a psychiatric emergency unit. This bias limits the generalization to other clinical settings or to the general population. Furthermore, this is a retrospective study, and this design limits the strength of the possible causal implications, particularly the temporal relationships between COVID-19 phases and psychopathological consequences. However, these limitations are also, at least in part, the strength of the study, as it focuses on a very severe subgroup of patients, who challenge the capacities of psychiatric management, being particularly sensitive to periods of crisis. Another strength of the study is the large number of outcome measurements. Despite the forgoing limitation, we consider our findings relevant because they derive from a consecutive, unselected sample of patients, without exclusion criteria, assessed with standardized measures, and with specific implications for routine diagnostic and treatment procedures.

## 5. Conclusions

In summary, the second lockdown represented a reason for greater concern, principally for self-harm ideation and behavior, both non-suicidal and suicidal, compared to the first lockdown. This was not associated with a greater rate of emotional or behavioral disorders, and it may be considered a marker of the persisting experience of COVID-19 and its related lockdowns, with growing effects, in a cumulative way. An additional reason of concern is that in times of enduring crisis, some clinical conditions may be partly excluded by the system of care, namely ADHD and intellectual disability. Finally, closer attention should be paid to the exposure to negative environmental conditions, particularly within the family.

## Figures and Tables

**Table 1 children-09-01921-t001:** Comparison between Pre-Lockdown (n = 115), First Lockdown (n = 65), and Second Lockdown (n = 61) groups: Categorical diagnoses (according to the DSM-5 criteria).

	Pre-Lockdown N = 115	First Lockdown N = 65	Second Lockdown N = 61	(df)	*p*	Comparisons between Groups
DSM-5 Diagnosis						
Depression, N (%)	6 (5.2)	6 (9.2)	3 (4.9)	1.4 (2)	ns	-
Bipolar disorder, N (%)	99 (86.1)	57 (87.7)	56 (93.4)	2.1 (2)	ns	-
Anxiety Disorders, N (%)	73 (62.9)	47 (72.3)	44 (72.1)	2.4 (2)	ns	-
ADHD, N (%)	38 (32.8)	28 (43.1)	10 (16.4)	10.6 (2)	0.005 *	FLG/SLG
ODD/CD, N (%)	49 (42.2)	36 (55.4)	29 (47.5)	2.9 (2)	ns	-
Autism Spectrum Disorder, N (%)	12 (10.3)	8 (12.3)	6 (9.9)	0.7 (2)	ns	-
Schizophrenia, N (%)	11 (9.6)	1 (1.5)	4 (6.6)	4.3 (2)	ns	-
Obsessive-Compulsive Dis., N (%)	14 (12.1)	8 (12.3)	8 (13.1)	0.04 (2)	ns	-
Tic/Tourette Syndrome, N (%)	3 (2.6)	4 (6.2)	2 (3.3)	1.5 (2)	ns	-
Post-Traumatic Stress Dis., N (%)	23 (19.8)	19 (29.2)	21 (34.4)	4.9 (2)	ns	-
Eating Disorders, N (%)	5 (4.3)	11 (16.9)	16 (26.2)	17.8 (2)	0.001 *	PLG/FLG PLG/SLG
Sleep Problems, N (%)	2 (1.7)	2 (3.1)	8 (13.1)	11.7 (2)	0.003 *	PLG/SLG
Substance Use Disorder, N (%)	15 (9.6)	11 (15.4)	5 (4.9)	3.9 (2)	ns	-
Tobacco use, N (%)	1 (0.9)	2 (3.1)	3 (4.9)	2.8 (2)	ns	-
Non-Suicidal Self-Injury, N (%)	35 (30.2	18; 27.7	32 (52.5)	10.9 (2)	0.004 *	PLG/SLG FLG/SLG
Persistent Suicide Ideation, N (%)	24 (20.7	10; 15.4	25 (41)	12.8 (2)	0.002 *	PLG/SLG FLG/SLG
Suicide Attempts, N (%)	13 (11.2	9; 13.8	18 (29.5)	10.3 (2)	0.006 *	PLG/SLG
Somatic Disorders, N (%)	3 (2.6)	1 (1.5)	5 (8.2)	4.7 (2)	ns	-
Learning Disability, N (%)	19 (16.4)	7 (10.8)	11 (18)	1.5 (2)	ns	-
Intellectual Disability N (%)	26 (22.6)	13 (20)	4 (6.6)	7.3 (2)	0.026 *	PLG/SLG

Legend: PLG: Pre-Lockdown Group; FLG: First Lockdown Group; SLG: Second Lockdown Group; ADHD: Attention Deficit Hyperactivity Disorder; ODD/CD: Oppositional Defiant Disorder, Conduct Disorder; * = *p* < 0.05; ns = not significant.

**Table 2 children-09-01921-t002:** Comparison between Pre-Lockdown (n = 115), First Lockdown (n = 65), and Second Lockdown (n = 61) groups, according to the Child Behavior Checklist (CBCL) scores.

CBCL-Scales, Mean (sd)	Pre-Lockdown; N = 115	First Lockdown; N = 65	Second Lockdown; N = 61	F (df)	*p*
Anxious/Depressed	72.3 (10.3)	69.8 (10.6)	71.6 (11.7)	0.9 (2)	ns
Withdrawn/Depressed	71.6 (11.4)	70.3 (12.1)	74.0 (14.1)	1.2 (2)	ns
Somatic complaints	63.6 (9.38)	63.0 (8.6)	64.8 (11.2)	0.4 (2)	ns
Social problems	67.4 (8.7)	66.2 (10.2)	65.3 (8.7)	0.8 (2)	ns
Thought problems	69.0 (9.2)	67.7 (8.4)	69.0 (10.5)	0.4 (2)	ns
Attention problems	68.5 (10.3)	68.1 (10.3)	64.1 (10.2)	2.8 (2)	ns
Rule-breaking behavior	65.1(10.5)	65.3 (9.6)	63.9 (9.2)	0.3 (2)	ns
Aggressive behavior	69.5 (12.3)	68.9 (10.3)	69.4 (12.3)	0.05 (2)	ns
Internalizing problems	71.0 (7.2)	69.1 (10.3)	71.4 (9.4)	1.1 (2)	ns
Externalizing problems	67.7 (9.8)	67.4 (9.8)	66.2 (10.7)	0.3 (2)	ns
Total problems	71.0 (6.6)	69.6 (8.6)	69.7 (8.4)	0.7 (2)	ns
ASEBA-Affective problems	74.8 (8.8)	71.6 (10.3)	75.7 (10.3)	2.6 (2)	ns
ASEBA-Anxiety Problems	69.5 (6.7)	67.9 (8.2)	67.9 (7.3)	1.1 (2)	ns
ASEBA-Somatic problems	59.9 (10.2)	60.4 (9.4)	61.8 (11.7)	0.5 (2)	ns
ASEBA- Attention Deficit-Hyperactivity problems	63.8(8.2)	64.7(8.5)	61.5 (7.9)	1.9 (2)	ns
ASEBA-Oppositional Defiant Problems	64.9 (9.2)	64.7 (8.4)	65.7 (9.6)	0.2 (2)	ns
ASEBA-Conduct problems	65.4 (11.1)	65.9 (9.6)	65.0 (10.2)	0.09 (2)	ns

Legend: CBCL; Child Behavior Checklist; ASEBA: Achenbach System of Empirically Based Assessment; *p* < 0.05; ns = not significant.

**Table 3 children-09-01921-t003:** Comparison between Pre-Lockdown (n = 115), First Lockdown (n = 65), and Second Lockdown (n = 61) groups: Direct and Indirect Adverse Childhood Experiences (ACE).

Direct ACE	Pre-Lockdown N = 115	First Lockdown N = 65	Second Lockdown N = 61	(df)	*p*	Comparisons between Groups
Sexual abuse, N (%)	0 (0)	2; 3.1	2; 3.3	3.7 (2)	ns	-
Physical maltreatment, N (%)	3 (2.6)	4; 6.2	1; 1.6	2.4 (2)	ns	-
Psychol. maltreatment, N (%)	3 (2.6)	3; 4.6	7; 11.5	6.3 (2)	0.043 *	PLG/SLG
Neglect, N (%)	9 (7.8)	12; 18.5	18; 29.5	14.3 (2)	0.001 *	PLG/SLG
**Indirect ACE**						
Witnessed violence, N (%)	7 (6)	8 (12.3)	5 (8.2)	2.2 (2)	ns	-
Alcoholism, N (%)	6 (5.2)	8 (12.3)	7 (11.5)	3.5 (2)	ns	-
Family member jailed, N (%)	5 (4.3)	2 (3.1)	1 (1.6)	0.9 (2)	ns	-
Uniparental family, N (%)	8 (6.9)	10 (15.4)	6 (9.8)	3.4 (2)	ns	-
Parental psychiatric dis., N (%)	7 (6)	6 (9.2)	14 (23)	11.9 (2)	0.003 *	PLG/SLG

Legend: ACE: Adverse Childhood Experiences; PLG: Pre-Lockdown Group; FLG: First Lockdown Group; SLG: Second Lockdown Group; * = *p* < 0.05; ns = not significant.

**Table 4 children-09-01921-t004:** Multiple linear regression.

	Overall %	B	df	Sig.	Exp (B)	95% CI Per EXP (B)	
						Lower	Upper
Groups (PLG)		**−0.668**	**1**	**ns**	0.513	0.109	2.415
Groups (FLG)		0.428	1	ns	1.534	0.321	7.332
Age		0.886	1	<0.001 **	2.426	1.491	3.949
Gender		−2.483	1	0.003 *	0.083	0.016	0.428
Indirect ACE: Witnessed violence		1.743	1	ns	5.713	0.520	62.729
Indirect ACE: Alcoholism		1.315	1	ns	3.725	0.147	94.538
Indirect ACE: Parental psychiatric disorder		1.786	1	0.044 *	5.965	1.050	33.883
Indirect ACE: Uniparental family		−2.148	1	ns	0.117	0.008	1.761
CBCL Internalizing problems		0.197	1	0.005 *	1.217	1.060	1.398
Constant		−20.315	1	<0.001 **	0.000	0.109	2.415

Legend: PLG: Pre-Lockdown group; FLG: First Lockdown group; ACE: Adverse Childhood Experiences; CBCL: Child Behavior Checklist; * = *p* < 0.05; ** = *p* < 0.01; ns = not significant.

## Data Availability

The data presented in this study are available upon request from the corresponding author.

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
