# Peer review of "How COVID-19 Phases Have Impacted Psychiatric Risk: A Retrospective Study in an Emergency Care Unit for Adolescents"

_children, 2022, doi:10.3390/children9121921_

Round 1

Reviewer 1 Report

Thank you for the opportunity to review this article. It touches on a very important and topical topic, but many things are missing or require improvement. On the formal side, the article is written with a lot of typos, which leaves a bad taste. The strength of this study lies in its application of a large number of outcome measures. However, the data analysis used was sparse, thus limiting the scientific soundness of this study. In my opinion, this article is a missed opportunity, as it stands, but it has potential. Therefore, I propose to deepen data analysis by creating regression models or structural equation modeling (SEM) analysis.

Detailed review:

1.       The language of this article leaves much to be desired. There were many unnecessary letters, dots, spaces, and commas. This study requires an in-depth proofreading service. The sadness is that the article contains 17 authors, none of whom found any sloppiness in preparing the manuscript for shipping. Honestly speaking, I perceive this as a disrespect for the reviewer and the journal. Correct me if I am wrong, but there is even a mistake on the list of authors.

2.       The introduction lacks the aim (s) of the work and hypotheses. In retrospective studies, a priori hypotheses should form the basis of the study, and in this study, it appears that the authors had the data and wanted to analyze it without knowing what they were looking for.

3.       The sample selection for this study was not explained. Participants were inpatients from the Psychiatric Emergency Unit of the Scientific Institute – the powerful Italian

4.       If I understand it correctly, 2014 is the approval date of the ethics committee. In turn, the study is from a project from 2021. What is the rationale for using the 8-year ethics committee approval issued under completely different conditions?

5.       Data analysis is poor. This is the usual intergroup comparison using the chi-squared test and analysis of variance. The strength of this study was the use of a large number of outcome measures. It would be useful to deepen these analyses, preferably by using regression models and, if possible, SEM/path analysis, and in the most extreme case, simple correlations.

6.       With such poor data analysis, it was difficult for me to respond to the discussion. However, there are certainly no limitations to this research, future study directions, or clinical implications.

7.       Unfortunately, the conclusion section does not summarize the results of the present study. Currently, it is a collection of free thoughts on the subject; however, it does not include the implications of this study.

8.       If I see it correctly, one author is missing from the authors’ contributions.

Reviewer 2 Report

A better understanding of the changes or possible increase in mental disorders after the first waves of Covid-19 is of main importance. This study compares the psychopathology present in three groups of patients who attended at a Psychiatric Emergency Unit. The first group was admitted before the first lockdown, a second one during the first lockdown and a third group (second lockdown). The results show that compared to those attended during the first lockdown the patients attended during the second lockdown present more psychological abuse and neglect, more psychiatric disorders in their parents, as well as suicidal ideation and behaviors and self-injurious not suicidal behavior.

It is an interesting article as it differentiates the post-pandemic time between the first and second lockdown and demonstrates a clearer worsening after the second. Factors such as the time of the pandemic or the situation/type of family (risk or protective factor) may be important in this situation, with notable changes in family and social relationships.

This is a retrospective study, where no clear hypotheses had previously been described to confirm or refute.

The topic is relevant, and the previous literature review is complete and current.

The manuscript is well written.The tables provide enough information. Some of the information displayed in tables can be removed from the text so as not to duplicate content (text referring to Table 2).

It is necessary to include a section on limitations of the study.

The conclusions, although they are wishes for better assistance, do not come directly from the study carried out.

Reviewer 3 Report

Thank you for an opportunity to review this article entitled "How COVID-19 phases have been impacting the psychiatric risk: a retrospective study in an emergency care unit for adolescents." Overall, I commend the authors for conducting this article. It sounds scientific, but need some closed attention to the details .

1. Introduction part should be more concise. Some unrelated issues could be deducted. You should focus on your research question, which is how COVID-19 impact the "diagnosis" and "clinical features" of psychiatric problems at the ED.

2. Table 1 should be extensively revised. It doesn't tell anything. What is C-GAS? It should be explained in text before existing in the table.

3. How you selected participants? It should be described more (i.e., randomly selected, convenience, cluster, etc.)

4. For 2.2 sub-heading, I don't know why the authors explain each score/scale. Please explain.

5. Again, Tables 2-3 should be extensively revised. Please remove "chi-square," "anova" column since I don't know what are the meanings.

6. Refs should not be cited in the Conclusion part as the authors need to conclude and summarize their important findings from the study and give further direction.

References should be typed in an MDPI format. Please kindly revise them.

Round 2

Reviewer 1 Report

The article has been improved and is now ready for publication. However, there are still quite a few imperfections in the preparation itself. Certainly limitation sectction should be moved from conclusion to discussion section. Also, many in-text citations have errors, separated by a semicolon instead of a comma. Also, authors' affiliation notes should be superscripted, there are differences in font types, not using the full journal template, etc. I understand that they will be corrected at the proofreading stage.

Reviewer 2 Report

I agree with the changes made.

Reviewer 3 Report

Several flaws are still needed to be revised.

1. Introduction is still not concise to me. Page 2, Paragraph 2 could be removed and paragraph 4 should be revised as they did not give more details regarding this study question.

2. You should not state "Results" in the methods section. You need to define what is your sample?, how your sample were derived?, what is your hospital or population settings? etc.

3. Table 1 should not be in the methods section and I'm not a statistician, as most of the readers, so you need to explain in the table whether what are the anova? and bonferroni-holme? Tables 1-5 should stand alone.

4. Tables contain a lot of flaws, for example, I don't think that 'ns' should be presented. You stated that *=p<0.05 why but in the tables, you type numbers and *. I don't see what are you want to communicate to readers.

5. You used many terms for describing diagnoses, conditions, etc. You should indicate in the methods which ref/resource do you retrieved this information.

6. The first paragraph in the conclusion is the limitations in this study. It should not be in that section.

4.

Round 3

Reviewer 3 Report

-